# A Distortion Correction Method Based on Actual Camera Imaging Principles

**DOI:** 10.3390/s24082406

**Published:** 2024-04-09

**Authors:** Wenxin Yin, Xizhe Zang, Lei Wu, Xuehe Zhang, Jie Zhao

**Affiliations:** State Key Laboratory of Robotics and System, Harbin Institute of Technology, Harbin 150001, China; zangxizhe@hit.edu.cn (X.Z.); wulei@stu.hit.edu.cn (L.W.); zhangxuehe@hit.edu.cn (X.Z.); jzhao@hit.edu.cn (J.Z.)

**Keywords:** camera calibration, lens distortion correction, image restoration, model-independent method

## Abstract

In the human–robot collaboration system, the high-precision distortion correction of the camera as an important sensor is a crucial prerequisite for accomplishing the task. The traditional correction process is to calculate the lens distortion with the camera model parameters or separately from the camera model. However, in the optimization process calculate with the camera model parameters, the mutual compensation between the parameters may lead to numerical instability, and the existing distortion correction methods separated from the camera model are difficult to ensure the accuracy of the correction. To address this problem, this study proposes a model-independent lens distortion correction method based on the image center area from the perspective of the actual camera lens distortion principle. The proposed method is based on the idea that the structured image preserves its ratios through perspective transformation, and uses the local image information in the central area of the image to correct the overall image. The experiments are verified from two cases of low distortion and high distortion under simulation and actual experiments. The experimental results show that the accuracy and stability of this method are better than other methods in training and testing results.

## 1. Introduction

In recent years, with the rapid development of computer vision and deep learning technology, cameras play an increasingly important role in human–robot collaboration sensing systems, especially in target detection and tracking and human motion recognition [1,2,3]. As a key technology of computer vision application, the accuracy and robustness of camera calibration directly affect the performance and reliability of robot sensing system [4]. The essence of camera calibration lies in establishing the mapping relationship between points in the scene and their corresponding points in the image, in which lens distortion is a problem that cannot be ignored. For instance, while the wide-angle camera can capture image data over a broad field of view, they are susceptible to issues such as significant deformation of edge pixels and serious information distortion. Therefore, in order to ensure the accuracy of subsequent algorithms, it is essential to correct the camera distortion.

Based on distinct calibration approaches, the camera calibration methods are typically classified into three categories: camera self-calibration method, active vision-based calibration method, and template-based calibration method. The camera self-calibration method utilizes the correspondence of the surroundings under multiple viewpoints during the camera’s natural motion to calibrate the reference target independently. Faugeras [5] and Maybank et al. [6] proposed a method based on Kruppa’s equation to solve the camera parameters directly. Liu [7] proposed a linear self-calibration method for projection cameras based on the representing of deforming object by the trajectory basis. The camera self-calibration method is excellently flexible, but has low accuracy, low robustness, and requires solving nonlinear equations. The camera parameters are determined through precise camera movement control in the active vision-based calibration method. De Ma [8] proposed a linear calibration method based on two sets of three orthogonal motions. Hartley [9] proposed a calibration method based on the pure rotational motion of the camera optical center. The active vision-based calibration method is simple and robust, but it has a high system cost, and is not applicable when the camera motion parameters are unknown or the camera motion cannot be precisely controlled. The template-based calibration method utilizes the known scene structure information to achieve camera calibration. Tsai [10] utilized a two-step calibration technique to achieve efficient calibration of the camera’s external position and attitude. Weng [11] and Gao [12] further investigated the two-step calibration. The method utilizes the metric information of three-dimensional [13], two-dimensional [14,15], or one-dimensional [16] templates to solve the camera model parameters with high calibration accuracy and robustness. Compared with the previous two methods, the template-based calibration method has wide applicability and provides more possibilities to solve the camera calibration problems in various practical applications. Taking into account the aforementioned factors, a template-based correction method was used in this study.

Currently, the most widely used camera calibration toolboxes [17,18] of the template-based calibration method are constructed based on Zhang’s technique [14] by capturing chessboard corner points. Heikkila [15] and Chen et al. [19] calibrated the camera by quadratic curve fitting based on circular array images without calculating the center of the circle. The above methods are based on the recognition results through iterative optimization to estimate the camera parameters and distortion coefficients simultaneously. However, this process leads to unstable parameter results due to the mutual compensation between parameters to reach the optimal solution. The analysis above indicates the need to correct the distortion individually and to obtain accurate camera parameters by correcting the control points.

In order to correct the distortion individually, several methods have been proposed to utilize the known structural information in the scene, such as straight lines, vanishing points, circles, etc. These methods aim to ascertain the undistorted position of the distorted points in the image. Methods [20,21,22,23] utilize the principle that straight lines in the scene remain straight in the image to correct the distortion. Xue [22] corrected the fisheye camera by learning the relationship between image distortion and linear curvature through a deep learning network. Liu [24] combined geometric constraints, such as vanishing point and ellipse, to self-calibrate lens distortion. The above methods exhibit limited noise immunity and stability during nonlinear optimization due to the use of non-metric information. Carlos [25] proposed a metric method for lens distortion correction based on globally structured images, which increases stability but results in low correction accuracy due to large distortions of the edge pixels. With the development of digital image technology, Xiong [26] and Weng et al. [27] determined the distortion displacement map of the entire image point-by-point based on the fringe pattern phase analysis method. Gao [28] and Jin et al. [29] computed the dense features between the original images and the projection of the reference images to obtain the distortion correction map based on the feature detection method of digital image correlation (DIC). Digital imaging techniques, while highly accurate in calibration, require pixel-by-pixel analysis and a significant amount of time for calibration. At the same time, shortening the camera’s working distance to maintain high accuracy results in losing information within the camera’s field of view. Rai [30] proposes an image correction method for C-arm system. The method maps a square near the image center to the grid to initialize the estimation of the distortion, and alternately updates the estimates of mapping and distortion by adding neighboring unmapped points in a breadth-first-search manner, and finally corrects the distortion individually. However, the multi-order distortion model in the iteration does not accurately fit the distortion, resulting in inaccurate distortion estimation. In addition, the initial estimation based on the four points in image center is susceptible to noise interference.

In this paper, a model-independent lens distortion correction method based on the image center area process is proposed, which is completed before the camera calibration. The method pre-corrects the pixel positions in the image from the principle of true camera distortion before calibrating the camera parameters. If a chessboard template is used, the distortion-corrected image can be used directly for pinhole camera calibration, avoiding the instability caused by mutual compensation of parameters during the optimization process. We propose the concept of image center area (ICA) and determine the size and location of the center area using perspective projection geometric invariants. The remaining points are optimized and extended based on the local image information of the determined center area to correct the distortion. The introduction of the image center area eliminates the effect of large distortions of the edge pixels, improves the accuracy of the correction, and shortens the optimization time.

The rest of the paper is organized as follows: Section 2 describes in detail the principles and methods of the proposed distortion correction, specifying how to correct distortion independently of the camera projection alone; Section 3 presents the results and analysis of the simulation and actual experiments; and finally, Section 4 draws conclusions.

## 2. Principles and Method

### 2.1. Geometric Constraints in Perspective Projection

The pinhole camera model, a widely applied camera imaging model, as shown in Figure 1a, describes the mapping of the point *P* in the world coordinate system to the point $P^{\prime }$ in the image coordinate system. The process in which point *P* is projected onto the image plane through the projection center is called perspective projection. The perspective projection essentially describes the transformation relationship between coordinate systems. The transformation does not preserve parallelism, length, and angle in the scene, but preserves collinearity and correlation. When the points in the scene are structured points, such as the chessboard template shown in Figure 1b, the points on the template are arranged in straight lines which are parallel, vertical, and equidistant from each other, the perspective projection result is shown in Figure 1c. The projection result preserves three projective geometric invariants: the crossing ratio between points remains constant [31], a straight line always remains a straight line [20,23], and parallel straight lines under projection intersect at a point, the vanishing point [32].

#### 2.1.1. Cross Ratio

Projective geometry preserves neither distances nor ratios of distances. However, cross ratios are preserved. As shown in Figure 1b,c, assuming that the corresponding pixel coordinates of the four points $P_{1},P_{2},P_{3},P_{4}$ in the chessboard template are $q_{1}=\left(u_{1},v_{1}\right),q_{2}=\left(u_{2},v_{2}\right),q_{3}=\left(u_{3},v_{3}\right)$ and $q_{4}=\left(u_{4},v_{4}\right)$, we obtain the following equation based on the invariant nature of the cross ratio:(1)$$F_{cr}\left(P_{1},P_{2},P_{3},P_{4}\right)=F_{cr}\left(q_{1},q_{2},q_{3},q_{4}\right)=\frac{d_{13}.d_{24}}{d_{14.}d_{23}}$$
where $F_{cr}$ denotes the function of the cross ratio, and $d_{ij}={\left(u_{i}-u_{j}\right)}^{2}+{\left(v_{i}-v_{j}\right)}^{2}$ represents the distance between $q_{i}$ and $q_{j}$.

The chessboard template is defined to consist of r straight lines, each line consisting of s points. In the distorted image, the distorted pixel $q_{m,n}^{d}=(u_{m,n}^{d},v_{m,n}^{d})$ is defined to represent the pixel coordinate of the nth point on the mth line, where *m* = 1, 2, … *r*, *n* = 1, 2, … *s*. To ensure that the distorted points satisfy the geometric constraints of the cross ratio in perspective projection, we should minimize the following error function:(2)$$E_{cr}=\sum_{m=1}^{r}\sum_{n=1}^{s-3}{\left(F_{cr}\left(q_{m,n}^{d},q_{m,n+1}^{d},q_{m,n+2}^{d},q_{m,n+3}^{d}\right)-F_{cr}\left(P_{1},P_{2},P_{3},P_{4}\right)\right)}^{2}$$

Each distorted pixel point $q_{m,n}^{d}$, as the intersection point of the horizontal and vertical lines, should satisfy the cross ratio of the horizontal and vertical directions at the same time. $F_{cr}\left(P_{1},P_{2},P_{3},P_{4}\right)$ is calculated in advance when designing the planar template.

#### 2.1.2. Straight Lines

In perspective projection, the geometric constraint that a straight line remains a straight line should be considered. In the distorted image, s points on the same line are not on a straight line because of the distortion, so it is necessary to fit these points to a straight line. The regression linear equation of the straight line $l_{m}$ is solved by the least squares. The equations are as follows:(3)$$\begin{array}{c} \hat{v}={\hat{a}}_{m}u+{\hat{b}}_{m}\\ {\hat{a}}_{m}=\frac{\sum_{n=1}^{s}\left(u_{m,n}^{d}-\overset{-}{u_{m}^{d}})(v_{m,n}^{d}-\overset{-}{v_{m}^{d}}\right)}{\sum_{n=1}^{s}{\left(u_{m,n}^{d}-\overset{-}{u_{m}^{d}}\right)}^{2}}\\ {\hat{b}}_{m}=\overset{-}{v_{m}^{d}}-{\hat{a}}_{m}\overset{-}{u_{m}^{d}} \end{array}$$
where $\left(\overset{-}{u_{m}^{d}},\overset{-}{v_{m}^{d}}\right)$ represents the average of s points on the line $l_{m}$, and $\hat{v}$ is the value of v corresponding to u after fitting the synthesized line.

To ensure that the distorted points in the perspective projection are on a straight line, it is necessary to ensure that the sum of the squares of the longitudinal distance from the distorted points to the straight line is minimized, i.e., minimize the following error function:(4)$$E_{st}=\sum_{m=1}^{r}\sum_{n=1}^{s}{\left(v_{m,n}^{d}-\left({\hat{a}}_{m}u_{m,n}^{d}+{\hat{b}}_{m}\right)\right)}^{2}$$

As previously mentioned, since the selected template has vertical and horizontal straight lines, Equation (4) will be minimized in both directions.

#### 2.1.3. Parallel Lines

As shown in Figure 1c, in perspective projection, the extension lines of the parallel lines in the image must intersect at a unique point, which is the vanishing point $q_{v}(u_{v},v_{v})$ located on the vanishing line. The chessboard template has two sets of parallel lines, horizontal and vertical, so there are two vanishing points. Based on the regression linear Equation (3), since all parallel lines intersect at one point, we can obtain the following equation:(5)$$\left[\begin{array}{cc} {-\hat{a}}_{1} & 1\\ {-\hat{a}}_{2} & 1\\ \ldots & \ldots \\ {-\hat{a}}_{r} & 1 \end{array}\right]\left[\begin{array}{c} u_{v}\\ v_{v} \end{array}\right]=\left[\begin{array}{c} {\hat{b}}_{1}\\ {\hat{b}}_{2}\\ \ldots \\ {\hat{b}}_{r} \end{array}\right]$$

Let Equation (5) be $A{\hat{q}}_{v}=b$, where ${\hat{q}}_{v}={\left[\begin{array}{cc} {\hat{u}}_{v} & {\hat{v}}_{v} \end{array}\right]}^{T}$ denotes the vanishing point. It is obtained by linear least squares as follows:(6)$${\hat{q}}_{v}={\left(A^{T}A\right)}^{-1}A^{T}b$$

If all parallel lines intersect at one point, the following error should be zero. Therefore, in order to ensure that the lines in the distorted image are parallel to each other, minimize the following error function to correct the distortion point:(7)$$E_{pa}=\sum_{m=1}^{r}{\left({\hat{b}}_{m}-\left({-\hat{a}}_{m}.{\hat{u}}_{v}+{\hat{v}}_{v}\right)\right)}^{2}$$
where based on the regression linear Equation (3), ${\hat{a}}_{m}$ and ${\hat{b}}_{m}$ are represented by distortion points $q_{m,n}^{d}=(u_{m,n}^{d},v_{m,n}^{d})$ in the image.

Similarly horizontal and vertical parallel lines need to be considered.

### 2.2. Image Center Area Projected through the Camera Lens

#### 2.2.1. Defining Image Center Area

In fact, modern cameras realize the imaging process through a complex combination of lens elements, rather than a hole. These elements are usually designed to be spherical [33] or aspherical. Due to the shape and optical properties of the spherical lens, the refraction of light through the spherical lens causes the deviation of the focusing position of the light from the ideal position, resulting in distortion, which is more obvious at the edge of the image. At the same time, when the light passes through the elements, the light at the edge region refracts at different angles of incidence than in the center region, so that more obvious distortion is generated at the edge the image. As shown in Figure 2a, it is assumed that the lens system is simplified to a single lens, when the angle of incidence $\theta_{2}>\theta_{1}$, the point $P_{2}^{\prime }$ has a larger offset than the point $P_{1}^{\prime }$ relative to the pinhole imaging result.

As shown in Figure 2b, when the angle relative to the principal axis is less than $\theta_{1}$, a circular region is formed in the image plane, which is defined as the image center area (ICA). The image within the ICA is closer to the image obtained from the pinhole model projection, and closer to the undistorted image, so we can use the local image information such as the green structured chessboard shown in Figure 2b to correct overall distortion.

#### 2.2.2. Determining Image Center Area

We use the reference grid to determine the location and size of the image center area. Firstly, the minimum reference grid is defined, which consists of 4×4 corner points. As shown in Figure 3, the set of 4×4 reference grids can be obtained by sliding the reference grid on the distorted chessboard image. Each minimum reference grid contains local error information E, which consists of straight-line error and cross-ratio error. As the reference grid is slid, the corresponding local error information is recorded for subsequent screening.

Next, the 4×4 grid set are initially screened. According to the characteristics of small distortion and small straight-line error within the ICA, we define the error of line $l_{m}$ combined with Equation (4):(8)$$E_{st}^{m}=\sum_{n=1}^{4}{{(v}_{m,n}^{d}-\left({\hat{a}}_{m}u_{m,n}^{d}+{\hat{b}}_{m}\right))}^{2}$$
where m=1,2,...,8 represents the eight lines of each minimum reference grid, and n represents the four points on each line. We sort the eight lines in each reference grid according to the $E_{st}^{m}$. Excluding the effect of large distortion of a single line, the average error of the middle six lines is taken as the straight-line error of the minimum reference grid:(9)$$E_{st}^{\prime }=\frac{1}{6}\sum_{m_{new}=1}^{6}E_{st}^{m_{new}}$$
where $m_{new}$ represents the middle six lines. According to the straight-line error $E_{st}^{\prime }$, all the reference grids are sorted, and the grids with small distortion are selected as the new grid set according to a certain proportion.

Due to the exclusion of the line with the largest linear error during the initial screening process, some reference grids may not be eliminated. For example, the reference grid 3 in Figure 3 has the smaller straight-line error with the exclusion of the upper line, but there is significant longitudinal distortion. Therefore, according to the principle that the closer the grid is to the center, the smaller the difference between the horizontal and vertical cross ratio is, i.e., the deformation along the two directions is the same, the cross-ratio error of the minimum reference grid is defined according to Equation (2):(10)$$E_{cr}^{l}=\sum_{m_{l}=1}^{4}{\left(F_{cr}\left(q_{m_{l},1}^{d},q_{m_{l},2}^{d},q_{m_{l},3}^{d},q_{m_{l},4}^{d}\right)-\frac{4}{3}\right)}^{2}\;E_{cr}^{v}=\sum_{m_{v}=1}^{4}\begin{array}{c} {\left(F_{cr}\left(q_{m_{v},1}^{d},q_{m_{v},2}^{d},q_{m_{v},3}^{d},q_{m_{v},4}^{d}\right)-\frac{4}{3}\right)}^{2} \end{array}\;E_{cr}^{\prime }=\left|E_{cr}^{l}-E_{cr}^{v}\right|$$
where $m_{l}$ represents the horizontal lines and $m_{v}$ represents the vertical lines. We get the final reference grid set using secondary screening based on the cross-ratio error $E_{cr}^{\prime }$. The minimum reference grid is a square, and there is a case where the deformation is the same along the two directions, such as the reference grid 2 in Figure 3. However, after excluding the line with the largest linear error, the straight-line error of the reference grid 2 is still very large, and it has been excluded in the initial screening process. Finally, according to the final reference grid set, the largest rectangular chessboard region contained within the grid set is taken as the image center area.

Considering the observation noise, if the number of chessboard points within the ICA is selected to be small, the stability is poor due to noise interference. If the number of chessboard points within the ICA is selected to be large, the convergence result is inaccurate due to the large deformation of the edge points, and the minimization time increases significantly with the increase of the number of variables. Based on the above considerations, for the size of the ICA, we need to maximize the number of chessboard points within the ICA and minimize the error of chessboard points within the ICA.

### 2.3. Distortion Pixel Correction

The distortion correction is essentially finding the undistorted pixel points corresponding to the observed points in the image. In order to obtain the undistorted pixel points in the image, combined with the three geometric constraints in the perspective projection, we define the following total error function *E*:(11)$$E={\lambda_{cr}E}_{cr}+\lambda_{st}E_{st}+{\lambda_{pa}E}_{pa}$$
where $\lambda_{cr},\lambda_{st},\lambda_{pa}$ are the weight coefficients of the three geometric constraint errors. Since the chessboard template contains horizontal and vertical lines, we must minimize Equation (10) in both directions. Using the Levenberg–Marquardt nonlinear minimization algorithm, the observed pixel points $\left(u_{d},v_{d}\right)$ in the original image are used as the initial values, and finally converge to the undistorted pixel points $\left(u,v\right)$ through multiple iterations. Compared with the existing calibration methods, which find the principal point, focal length, rotation, and translation vector parameters through nonlinear minimization iteration, nonlinear minimization to find pixel point coordinates has better conditions. Because when any point is changed by one unit, the change of the error function value produced by all variables is equal.

Equation (10) consists of three error functions: cross-ratio error, straight-line error, and parallel error. Although the final result of each error function converges to zero, the inconsistency of the dimension of each error function leads to different convergence speeds, which has a great impact on the final optimization result and optimization speed. Therefore, we have to consider adding weight coefficients to each function. We optimize each error function individually, and set the order of magnitude of the error function as the benchmark weight when it is close to convergence. Considering that the parallel error is calculated based on the straight lines, the convergence speed of the parallel error is faster than that of the straight-line error, and the straight-line error also affects the parallel error, so the weight of the parallel error should be reduced appropriately.

We consider the point set obtained by minimizing the correction of Equation (10) to be the “correct point set” because it satisfies all the geometric constraints under the perspective transformation. However, in fact, since the distance between points is represented by the cross ratio rather than a fixed size, there are many sets of points that satisfy the constraints of the chessboard template. Since the green chessboard region within the ICA is closer to the undistorted image, we only correct the positions of the chessboard pixel points within the ICA. Finally, the remaining distorted points in the image are extended by the previously defined geometric constraints, and the “best undistorted point set” of the image is calculated to ensure uniqueness.

### 2.4. Degenerate Configurations

It can be seen from the proposed method that when the camera is located directly above the calibration template plane, the lines in the captured chessboard image are parallel to each other. In this case, the parallel lines do not intersect, resulting in the lack of parallel constraints in the process of correcting. Therefore, it is important to ensure that the camera is tilted with the template plane to avoid degraded configurations when taking pictures.

For the chessboard image with large distortion, it is more accurate to determine the image center area using a reference grid. However, when the distortion is small and the observation noise is large, the offset caused by the noise is larger than the distortion offset. At this time, it is not accurate to determine the image center area using a reference grid. We can only select chessboard points directly in the center of the image to represent the image center area.

At the same time, the quantity and quality of the data in the image is more important to ensure the accuracy of the correction, where the data quantity refers to the number of control points and the data quality refers to the distribution of control points. Firstly, as shown in Figure 4a, the fewer the number of control points, the fewer the control points within the ICA, and the larger the error after expansion. Therefore, more control points are needed to ensure the number of chessboard points within the ICA, so as to ensure the accuracy of calibration. Secondly, the control points must be projected in the entire image to characterize the lens distortion more accurately and comprehensively. Because the distortion is more easily observed in the boundary of the image, as shown in Figure 4b, when the control points are concentrated in the center area, the lens distortion will be only partially represented, leading to inaccurate subsequent parameter calculations. As shown in Figure 4c, the quantity and quality of the data are guaranteed, so as to ensure the accuracy of the correction.

## 3. Experiments and Results

In this experiment, three different lens distortion correction techniques are used as a comparison to correct the chessboard image. The first is the overall calibration method of Zhang [14]. Firstly, assuming that the image is undistorted, the initial values of the pinhole model coefficients are obtained by the one-to-one correspondence between the obtained image points and the template plane points. Then, assuming that the obtained pinhole model coefficients have no error, the initial values of the distortion coefficients are obtained by the undistorted points derived from the world coordinate and the distorted points in the image. Finally, the above coefficients are optimized by using the full camera model, and the pinhole model coefficients and distortion coefficients are derived by mutual compensation.

The second method Is the metric nonlinear lens distortion correction method proposed by Carlos [25]. The method first utilizes all the structured metric information in the image, including the cross ratio and the straight line to globally correct the image. Secondly, assuming that the distortion center is at the center of the image, the nonlinear optimization technique is used to optimize the distortion center and the distortion coefficients.

The last method uses the distortion correction method based on the image center area introduced in this paper. In this paper, the global correction of the image is performed using local structuring information, and the pinhole model coefficients and distortion coefficients are calculated by using the correction information.

We conducted experiments on computer simulated data and actual data. In the experiment, 15 images of the same chessboard template were obtained from different perspectives, and 10 of them were corrected by the above three methods to verify the accuracy and stability of different distortion correction methods. Next, the polynomial model composed of radial and eccentric distortion proposed by Brown [34,35] was used to obtain the distortion coefficients of the different methods. Then, the remaining five images were corrected based on the distortion coefficients to verify the reliability of the camera parameters obtained using different methods. Both simulation and actual experiments verify the effectiveness of the method from the two distortion cases of normal low distortion and fisheye high distortion.

### 3.1. Evaluation Indicators

In order to evaluate the accuracy of the distortion correction methods and to investigate the influence of different factors on the distortion correction results, intuitively, we can measure the distance between the undistorted real point $q_{i}=(u_{i},v_{i})$ projected onto the image plane by the simulation camera and the corrected point $q_{i}^{*}=(u_{i}^{*},v_{i}^{*})$:(12)$$e_{d}=\frac{1}{n}\sum_{i=1}^{n}\sqrt{{\left(u_{i}^{*}-u_{i}\right)}^{2}+{\left(v_{i}^{*}-v_{i}\right)}^{2}}$$

The calibration error can also be defined in terms of the distance between the 3D point $p_{i}=(x_{i},y_{i},z_{i})$ in the camera coordinate and the 3D point obtained by back-projecting the corrected point into the camera plane. These differences can be visualized. However, they are sensitive to the digital image resolution, camera field of view, and object-to-camera distance. The normalized calibration error (NCE) proposed by Weng [11] overcomes this sensitivity with the following equation:(13)$$e_{nce}=\frac{1}{n}\sum_{i=1}^{n}\sqrt{\frac{{\left(x_{i}^{*}-x_{i}\right)}^{2}+{\left(y_{i}^{*}-y_{i}\right)}^{2}}{z_{i}\left({f_{x}}^{-2}+{f_{y}}^{-2}\right)/12}}$$
where $p_{i}^{*}=(x_{i}^{*},y_{i}^{*},z_{i})$ is the camera coordinates of the 3D point obtained by back-projecting the corrected pixel point $q_{i}^{*}$ to depth $z_{i}$, $f_{x}$ is the row focal length of the camera, and $f_{y}$ is the column focal length of the camera. If $e_{nce}<1$, it means that the calibration error is lower than the digital noise of the pixel at this depth. If $e_{nce}=1$, it indicates a good calibration, and the residual distortion is negligible compared to the image digitization noise at this depth. $e_{nce}>1$ reveals poor calibration.

We reproject the 3D points into the 2D image through the pinhole and distortion models, and the distance between the reprojected points and the corresponding observation points in the image is used as the evaluation metric, also known as the reprojection error.

### 3.2. Computer Simulated Data

In the computer simulation, we use the pinhole model as the virtual camera imaging model, with the parameters: row focal length $f_{x}=1000$, column focal length $f_{y}=1000$, principal point $\left(u_{0},v_{0})=(960,540\right)$, and the camera resolution of 1920×1080. The template plane is a chessboard pattern containing 19×12=228 corner points, with adjacent corner points spaced 50 mm apart. The pattern size is 950 mm×600 mm. We add low distortion and high distortion to the camera normalization plane based on the mathematical model of distortion. The virtual camera obtains 15 images from different perspectives while ensuring that the chessboard fills as much of the image as possible. Ten images were used to correct and compute the coefficients, and five images were tested based on the distortion coefficients, where the test image data did not add noise.

#### 3.2.1. Correction Accuracy and Image Center Area Size

In order to investigate the effect of the size of ICA on the correction accuracy, in this experiment, we corrected the plane with the rotation vector $r={\left[10{}^{\circ},0{}^{\circ},1{}^{\circ}\right]}^{T}$ and the translation vector $t={\left[-450,-275,600\right]}^{T}$ multiple times by changing the size of ICA. Since the same plane is used and the direction and the size of the chessboard are unchanged, we can use the number of chessboard points within the ICA to represent the size of ICA. We add Gaussian noise with a mean value of 0 and σ=0.2 to the image. For different sizes of ICA, 10 independent experiments were conducted, respectively, and the image pixel error is expressed more intuitively using Equation (12). The results are shown in Figure 5.

Figure 5a,b show the correction errors under normal low distortion and fisheye high distortion, respectively. When the image center area is small, the number of chessboard points within the ICA is small, which is greatly affected by noise and has poor stability, resulting in a large correction error obtained using the expansion of the central area. When the image center area is large, the number of chessboard points within the ICA is large and the stability is strong. However, due to the large deformation of the edge points, as shown by the blue line in the figure, the correction error of the points within the ICA increases consequently, leading to an increase in the overall correction error. For the plane in this experiment, when there are about 50 corner points within the ICA (the internal chessboard size is about 9×6), the distortion correction error is the smallest because the number of chessboard points is sufficient and closer to the undistorted points. This also proves that we need to maximize the number of chessboard points within the ICA and minimize the chessboard point error within the ICA to achieve the best correction effect. At the same time, as shown in Figure 5c, although the optimization time increases with the number of points, it is negligible when the number of points is less than 60. When the number of points within the ICA is 228, which contains all the points in the chessboard template, it is the same as the Carlos global correction. At this time, the optimization time is the longest, and the error is also large.

#### 3.2.2. Correction Accuracy and Pose and Position of the Template

In this experiment, we investigated the effect of the pose and position of the chessboard template on the correction accuracy. The pose is represented by the rotation vector between the camera coordinate system and the world coordinate system where the template plane is located, and the position is represented by the translation vector.

Firstly, the effect of the pose is discussed. Define the rotation vector of the template plane as $r={\left[\theta ,0{}^{\circ},1{}^{\circ}\right]}^{T}$, where θ gradually increases from 0° to 60° with the step size of 5°. At this time, the angle between the camera plane and the template plane is approximately θ. The distance between the center of the template and the camera remains constant. Gaussian noise with a mean value of 0 and σ=0.2 is added to the image. We found the optimal size of ICA corresponding to each angle, and conducted 10 independent experiments based on the optimal image center area. The image pixel error is expressed more intuitively with Equation (12).

The results are shown In Figure 6a,b, where the pixel correction error decreases with the increase of the angle for both low and high distortion. When the distance between the template and the camera remains constant, because the actual size of the chessboard does not change, the size of the chessboard as presented in the image will reduce with the increase of the angle, leading to a decrease in the number of pixels that the projected chessboard occupies. As shown by the red lines in Figure 6a,b, the overall distortion decreases with the increase of the angle. At the same time, as shown in Figure 6c, the number of points within the optimal image center area increases with the increase of the angle, indicating that more points close to the undistorted points can be used to correct, resulting in a smaller error. The variation of the angle of the template mainly affects the size of the chessboard in the image, independent of our proposed method. Our method corrects the distorted image significantly under different angle conditions. In practice, we should choose a smaller angle to ensure that the chessboard covers the entire image, so as to completely characterize the lens distortion of the camera.

As shown in Figure 7, although the size of the chessboard template’s projection in the image decreases with the increase of the angle, the size and position of the optimal ICA remain almost constant, approximately located in the center of the image and accounting for about 20% of the image plane. At the same time, as shown in Figure 6c, the smaller the distortion is, the larger the ICA size is. The number of corner points within the ICA is primarily determined by the distribution of chessboard points; if the distribution is dense, the number will be more.

Secondly, the effect of the position is discussed. We corrected the plane with the rotation vector $r={\left[10{}^{\circ},0{}^{\circ},1{}^{\circ}\right]}^{T}$ and the translation vector $t={\left[x,-275,600\right]}^{T}$, where x gradually increases from −330 to −570 with the step size of 30 to represent the horizontal movement of the chessboard projected to the image. Gaussian noise with a mean value of 0 and σ=0.2 is added to the image.

The results are shown in Figure 8, where the x-axis is represented by the value of the horizontal coordinate of chessboard center. When the value is 960, the chessboard is in the horizontal center of the image. When the chessboard is located at the edge, the error is greater than when the chessboard is located at the center. However, this variation is not very pronounced. The correction error varies with the overall distortion, independent of our proposed method. It is important to ensure that the chessboard fills the image center area to improve the correction accuracy.

Above, we assume that the distance between the template and the camera is constant. As for the effect of the distance on image pixel error, as the distance increases, the number of pixels that the projected chessboard occupies in the image decreases, resulting in a smaller size of the chessboard in the image with less distortion, which is similar to the effect of the angle.

#### 3.2.3. Correction Accuracy and Pixel Noise

In practice, as the distance between the template and the camera increases, the projected size of the template on the image becomes relatively smaller. This reduction in size leads to a decrease in the amount of information contained within each pixel, which in turn reduces the resolution of image details and may lead to a decline in the precision of corner detection. The chessboard corner detection algorithms, such as the Harris method [36], exhibit varying levels of sensitivity to changes in the angle of the template plane, which can potentially affect the accuracy of corner detection. Moreover, the printing accuracy affects the printing quality of the picture, including the clarity, detail, and noise level of the picture, which also affects the detection accuracy of the corner. To assess the effect of corner detection precision on correction accuracy, we introduce pixel noise into the imaging process for consideration.

In this experiment, we investigated the effect of Gaussian noise on the correction accuracy. The position and pose of the template do not affect the effectiveness of our method, but they do affect the overall distortion of the image. We captured 15 images, each of which filled the chessboard as much as possible to ensure that each image had a similar distortion. The Gaussian noise was gradually increased from 0.1 to 1.0, and the step size was set to 0.1 with a total of 10 noise levels. The correction accuracy and the test accuracy of the three methods is compared by the normalized calibration error.

As shown in Figure 9, with the increase of noise, the accuracy of Zhang’s method and proposed method decrease, and the Carlos’s method is relatively stable. Because the Carlos’s method is a global correction, its correction result tends to be stable when the effect of distortion is greater than the noise. However, the distorted points far away from the distortion center directly affects the global correction result due to the large distortion, resulting in the largest correction and test errors among the three methods. In Zhang’s method, the distortion coefficients and pinhole parameters are first calculated simultaneously by the model, and then are used for image correction. The proposed method directly corrects the image independent of the model, so the proposed method is superior to Zhang’s method in the distortion correction. However, when testing based on the distortion coefficients, the current mathematical model cannot fit the distortion well, resulting in the testing error of the proposed method being larger than the direct correction error. The test results are optimal except for the reduced test accuracy for large noise at low distortion. Overall, our method shows good results in terms of anti-noise performance.

Figure 10 shows the simulated chessboard image correction results of the three methods when the pixel noise is 0.2. It can be seen from the Figure that the farther the pixel is from the distortion center, the greater the distortion is, and the more unstable the correction result is. The results of Zhang are expanded relative to the true results, the results of Carlos are shrunk relative to the true results, and the proposed method is the closest to the real results.

#### 3.2.4. Correction Accuracy and Number of Points

In this experiment, we investigated the effect of the number of points in the chessboard template on the correction accuracy. We add Gaussian noise with a mean value of 0 and σ=0.2 to the image. As shown in Figure 11, as the number of points increases, the correction accuracy of all methods increases. The proposed method has the smallest error in both the correction and testing process, and when the number of points in the chessboard template is greater than 500, the error tends to be stable.

### 3.3. Actual Data

We used two hardware devices to verify the accuracy of the distortion correction method. The first is the Kinect from Microsoft, representing the low distortion case. The second is the Canon’s fisheye camera, representing the high distortion case. Each camera takes 15 images of the chessboard template from different perspectives with 1920×1080 pixels. In order to ensure that the distortion in each area of the image can be represented, the chessboard needs to be filled with the entire image as much as possible, while minimizing the effect of different distortion due to the different imaging sizes of the checkerboard. Ten images were used for direct correction and calculating the coefficients, and five images were used for testing.

#### 3.3.1. Direct Correction Results

Figure 12a shows a chessboard template image with high distortion obtained using the fisheye camera. Figure 12b shows the point set corrected using Zhang’s method. Figure 12c shows the point set globally corrected using Carlos’s method. Figure 12d shows the point set corrected using the proposed method based on the image center area. The correction results of the latter two methods satisfy the geometric constraints described in the principle, and the undistorted points after global correction look closer to the distorted points. However, the deformation of the global correction does not increase from the image center to the image edge, and the direction of the deformation in the center is opposite to the direction of the deformation at the edge, which does not conform to the principle of camera lens distortion. Zhang’s method utilizes the distortion mathematical model to correct the distortion. However, because the model cannot fit the distortion well, the straight lines appear curved after correction, which does not satisfy the fact that a straight line remains straight in perspective projection. The results of the proposed method show that the undistorted points in the center area are closer to the distorted points, and the farther the point is from the center, the greater the deformation is, which is consistent with the camera lens distortion principle and closer to the correct undistorted points. This is similar to the simulation results corresponding to Figure 10.

#### 3.3.2. Correction Results Based on Distortion Model Coefficients

As shown in Table 1, we first calculate the pinhole parameters and distortion coefficients of each method and test based on these parameters, where the errors are represented by reprojection error. In Zhang’s method, the pinhole parameters and distortion coefficients are calculated simultaneously by minimizing the reprojection error using a nonlinear optimization method. To satisfy the training dataset, the parameters are compensated with each other, which works well, but large errors occur when testing with new images. The calibration parameters of Zhang do not really represent lens distortion and pinhole camera. The correction results of Carlos’s method cannot represent the lens distortion well, which indirectly affects other parameter estimation, so the training error and test error are larger. Compared with the method of Zhang and Carlos, the proposed method has better results by calculating the camera model and the distortion model separately while ensuring the accuracy of the distortion correction. As shown in Figure 13, the correction results of different method based on the distortion model coefficients is similar to the simulation results and the direct correction results.

## 4. Conclusions

In this paper, we have proposed a model-independent lens distortion correction method based on the image center area. This method is based on the idea that the structured image preserves their ratios through perspective transformation. These ratios are independent of the position, orientation, and characteristics of the camera that captured the image, and relate only to the calibration template. At the same time, we propose the concept of image center area (ICA), which uses the local image information in the central area of the image to optimize and extend to obtain the remaining points to correct the distortion. After determining the undistorted points, the camera parameters and distortion coefficients are directly calculated by using the known pinhole model and distortion model, which avoids the parameter optimization process and reduces the time consumption.

The experiments are verified from two cases of low distortion and high distortion under simulation and actual experiment, comparing the correction errors of the training images and the test images in different methods, where the test images are corrected based on the model parameters. The experimental results show that the distortion correction errors using local information is smaller than Carlos’ global distortion correction method, while the optimization time is greatly reduced. Compared with Zhang’s method, calculating the camera parameters and distortion coefficients separately avoids the instability caused by the simultaneous optimization of the parameters and achieves better results.

We considered further related work. The experimental results indicate that the correction accuracy of the proposed method, when the optimal image center area is established, is independent of the size, pose, and position of the template, and is primarily related to the quality of the image. To enhance the precision of the correction, it is essential to conduct a thorough study on accurate corner detection. Secondly, since none of the current mathematical distortion models can fit the actual distortion well, we can obtain the lens distortion corresponding to each pixel through pixel interpolation directly or indirectly. Finally, we can apply this method from single-camera calibration to multi-camera calibration.

## Figures and Tables

**Figure 1 sensors-24-02406-f001:**
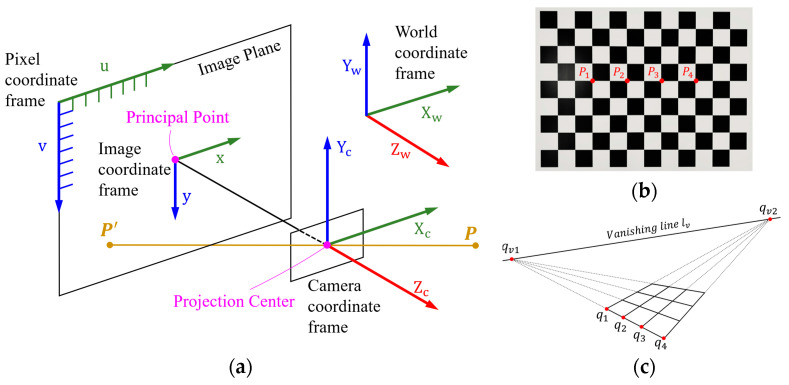
(**a**) Pinhole camera model; (**b**) chessboard used as template; (**c**) in the perspective projection, the cross ratio does not change. Straight lines remain straight. Parallel lines intersect at the vanishing point.

**Figure 2 sensors-24-02406-f002:**
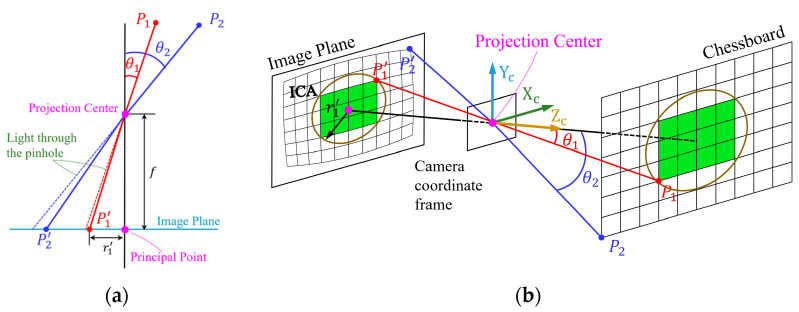
(**a**) Distortion increases with the increase of the angle of the beam passing through the camera lens; (**b**) chessboard template imaging through the camera lens. The green area in the image plane represents the chessboard in the ICA, which is closer to the undistorted chessboard image.

**Figure 3 sensors-24-02406-f003:**
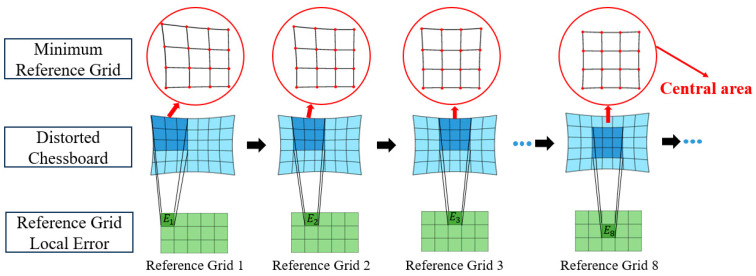
The sliding process of the minimum reference grid on the distorted chessboard image.

**Figure 4 sensors-24-02406-f004:**
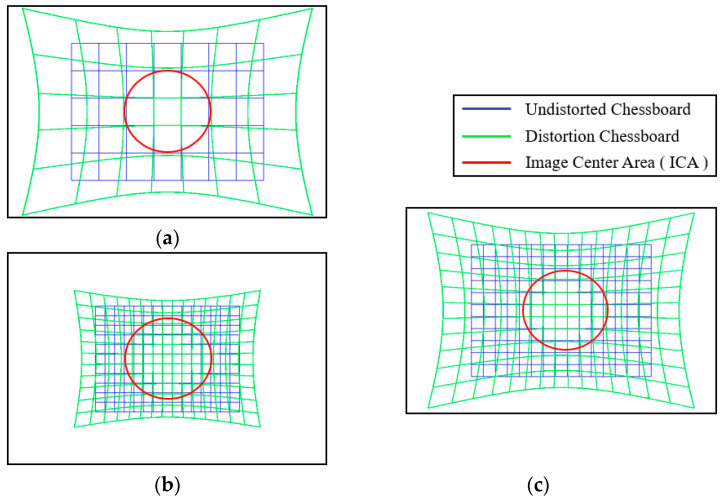
(**a**) The small amount of data leads to less control points within ICA; (**b**) the concentration of data in the center area causes lens distortion to be partially represented; (**c**) sufficient data and projection of the chessboard over the entire image ensure correction accuracy.

**Figure 5 sensors-24-02406-f005:**
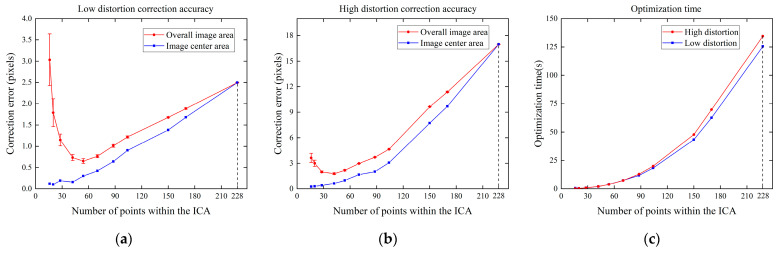
(**a**) Effect of the size of ICA on correction accuracy at low distortion; (**b**) effect of the size of ICA on correction accuracy at high distortion; (**c**) effect of the size of ICA on optimization time.

**Figure 6 sensors-24-02406-f006:**
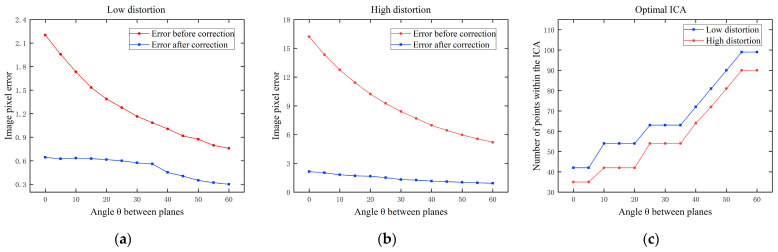
(**a**) Effect of the angle on image pixel error at low distortion; (**b**) effect of the angle on image pixel error at high distortion; (**c**) effect of the angle on number of points within the optimal image center area.

**Figure 7 sensors-24-02406-f007:**
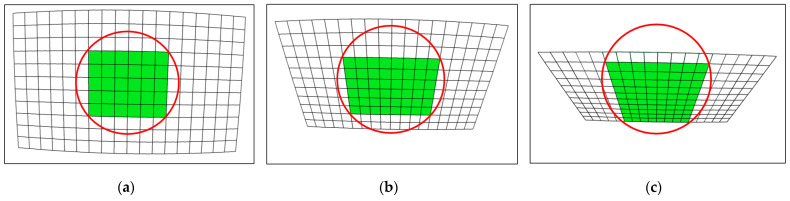
Optimal image center area at different angles where ICA is represented by the red circle: (**a**) θ=5°; (**b**) θ=30°; (**c**) θ=55°.

**Figure 8 sensors-24-02406-f008:**
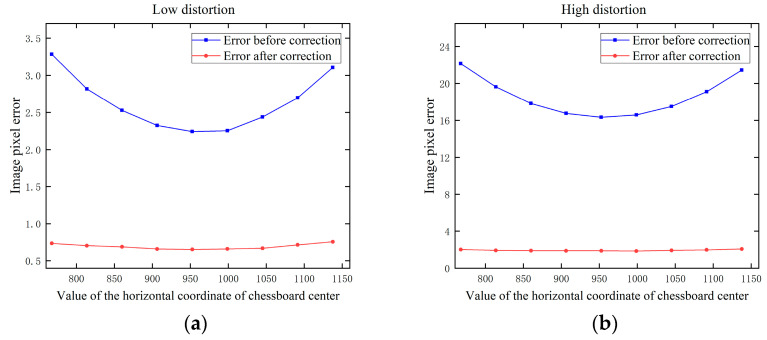
Effect of the position of the chessboard on image pixel error: (**a**) low distortion; (**b**) high distortion.

**Figure 9 sensors-24-02406-f009:**
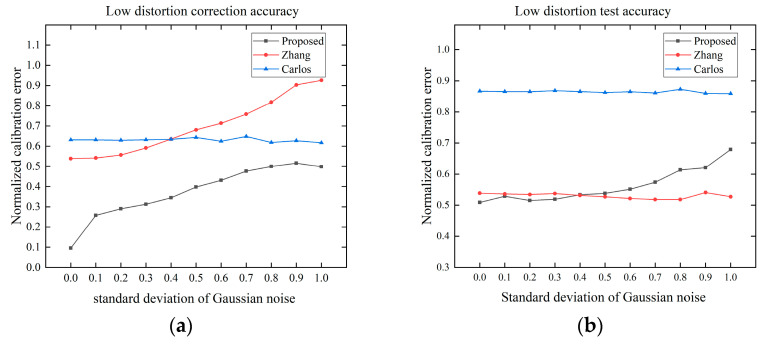
Effect of the pixel noise on correction accuracy: (**a**) low distortion correction accuracy; (**b**) low distortion test accuracy based on the distortion coefficients; (**c**) high distortion correction accuracy; (**d**) high distortion test accuracy based on the distortion coefficients.

**Figure 10 sensors-24-02406-f010:**
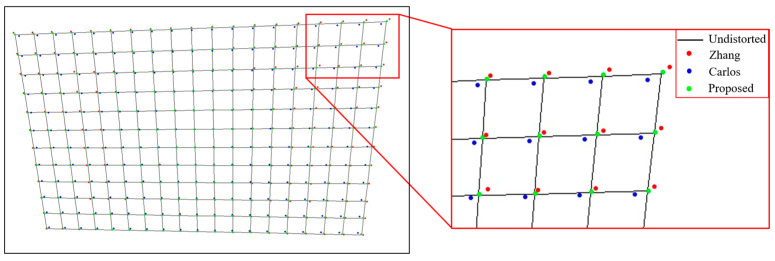
Simulated chessboard correction results. The farther the pixel is from the center of the image, the more unstable the distortion correction is.

**Figure 11 sensors-24-02406-f011:**
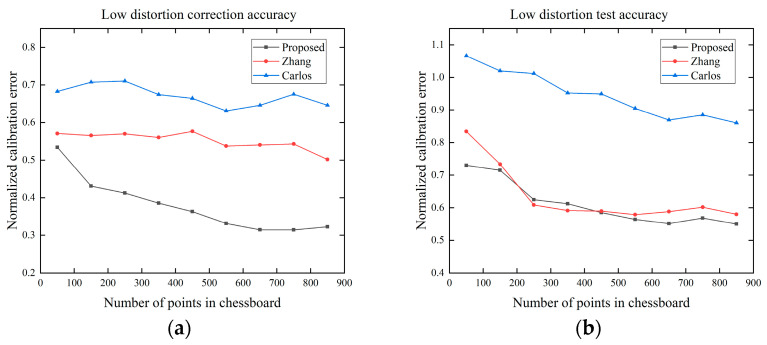
Effect of the number of points in chessboard on correction accuracy: (**a**) low distortion correction accuracy; (**b**) low distortion test accuracy based on the distortion coefficients; (**c**) high distortion correction accuracy; (**d**) high distortion test accuracy based on the distortion coefficients.

**Figure 12 sensors-24-02406-f012:**
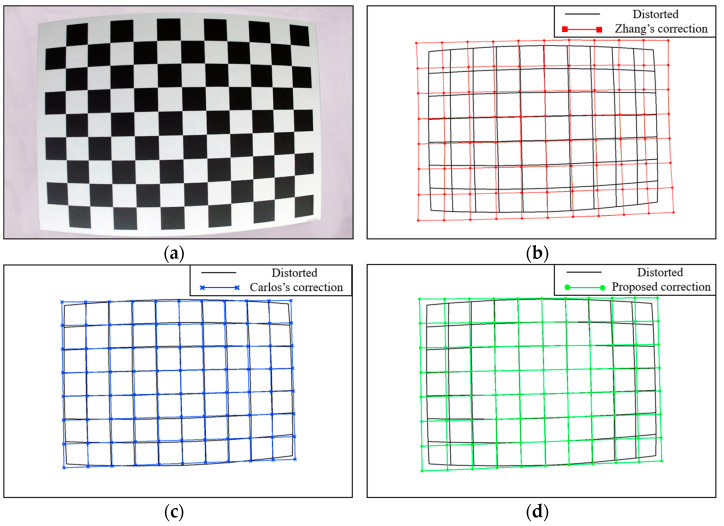
(**a**) The pattern of the chessboard template with high distortion; (**b**) point set corrected using Zhang’s method; (**c**) point set globally corrected using Carlos’s method; (**d**) point set corrected using the proposed method based on the image center area.

**Figure 13 sensors-24-02406-f013:**
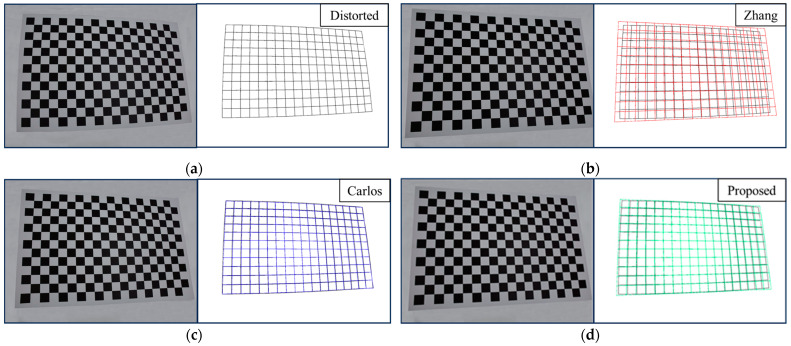
Corrected patterns based on distortion model coefficients calculated using different methods: (**a**) original distortion pattern; (**b**) the pattern corrected using Zhang’s coefficients; (**c**) the pattern corrected using Carlos ‘s coefficients; (**d**) the pattern corrected using coefficients of the proposed method.

**Table 1 sensors-24-02406-t001:** Pinhole parameters and distortion coefficients calculated by different methods, where $k_{1}$ and $k_{2}$ are radial distortion coefficients, and $p_{1}$ and $p_{2}$ are tangential distortion coefficients.

Parameters	Kinect			Canon		
	Zhang	Carlos	Proposed	Zhang	Carlos	Proposed
$f_{x}$	1046.11	1052.20	1058.92	1318.68	1171.95	1312.92
$f_{y}$	1046.62	1051.28	1058.13	1317.91	1175.95	1318.71
$u_{0}$	931.68	931.18	931.66	951.95	968.584	951.01
$v_{0}$	554.33	554.45	556.11	543.68	551.169	553.03
$k_{1}$	0.0498	−0.0092	0.0072	−0.2439	0.1119	−0.0977
$k_{2}$	−0.0550	0.0389	0.01576	0.1443	−0.5595	−0.2225
$p_{1}$	-	−0.00004	0.00015	-	−0.00015	0.00081
$p_{2}$	-	−0.00031	-0.00029	-	−0.00084	−0.00063
Training reprojection error	0.393	1.083	0.912	0.336	3.392	2.071
Testing reprojection error	1.714	1.536	1.092	3.535	3.827	2.914

## Data Availability

The data presented in this study are available on request from the corresponding author.

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
