# Peer review of "A Distortion Correction Method Based on Actual Camera Imaging Principles"

_sensors, 2024, doi:10.3390/s24082406_

Round 1
Reviewer 1 Report
Comments and Suggestions for Authors
1.Please supplement the analysis of the impact of the printing accuracy of the target pattern and the camera installation position on the accuracy of distortion correction,please try to give out the mathematical relationship between distortion correction accuracy and target printing error, imaging distance, optical Angle resolution and other relevant parameters.
2.This method has high application value in engineering, especially in batch development of high precision star sensor and small optical measurement load
Reviewer 2 Report
Comments and Suggestions for Authors
in attached file

Reviewer 3 Report
Comments and Suggestions for Authors
In this paper, a distortion correction method based on the projective transformation property and invariants is proposed. The proposed method is based on the idea that the structured image preserves its ratios through perspective transformation, and uses the local image information in the central area of the image to correct the overall image. And the experiment results show that the proposed method had better performance in cutting down the distortion correction error.
However, I think this paper is more like a method about camera calibration. For most camera calibration, the distortion correction is performed after the camera is calibrated. For carlos and this paper, the distortion correction is performed at first then followed calibration. For carlos, he corrected the calibration image based on global information, while the proposed method the image is corrected based on ICA information. But the idea of conducting distortion correction using central area information before pin-hole calibration has been carried out in XRII C-arm image correction and calibration. One contribution of this paper is that both the position and size of the ICA region in the proposed method is dynamically determined by an evaluation method.
Other problems:
1. In Fig.1(a), the coordinate is not right-handed. Generally, right-handed coordinate is preferred.
2. In Line 139, mth and nth, should be m^{th} and n^{th}.
3. In Eq.3, line expression, v=bu+a, why not use y=ax+b or v=au+b, a more familiar expression? And further more, why there is a hat for v, but not hat for u?
4. Eq.7 is an object function for optimization, but what are the variants to be optimized?
5. In Eq.13, 'alpha and beta are the intrinsic parameters of the camera', is not rigorous.
6. In Fig.6, subcation for subfigure a, b, c, and d should be added directly to distinguish these subfigure.
Comments on the Quality of English Language
The quality of the english language is just ok. It should be improved for better understanding of the readers.
Round 2
Reviewer 2 Report
Comments and Suggestions for Authors
I admire the work of the authors to the improvement of the paper, the quality is now much better. The authors responded sufficiently to all my comments. My only minor note:
lines 188-190 I suggest reformulating as: the light at the edge region refracts at different angles of incidence than in the center region, so that more obvious distortion…..
it is not true that an additional refraction may occur
Author Response
Thank you very much for providing important improvement suggestions for the paper from the perspective of an optical engineer. We highly value your professional insights and have already made modifications to the paper based on your recommendations.
Point: lines 188-190 I suggest reformulating as: the light at the edge region refracts at different angles of incidence than in the center region, so that more obvious distortion…..
Response: Thank you for your suggestion. I have rephrased lines 188-190 as you suggested.
At the same time, when the light passes through the elements, the light at the edge region refracts at different angles of incidence than in the center region, so that more obvious distortion is generated at the edge the image.